# The Impacts of a Large Water Transfer Project on a Waterbird Community in the Receiving Dam: A Case Study of Miyun Reservoir, China

**Waner Liang** [1,2,†], **Jialin Lei** [1,2,†], **Bingshu Ren** [3], **Ranxing Cao** [1], **Zhixu Yang** [1], **Niri Wu** [1] and **Yifei Jia** [1,2,*]

1   School of Ecology and Nature Conservation, Beijing Forestry University, Beijing 100083, China;
    vandy0303@bjfu.edu.cn (W.L.); leijialinbjfu@bjfu.edu.cn (J.L.); ranxingcao@bjfu.edu.cn (R.C.);
    yang_zhixu0119@bjfu.edu.cn (Z.Y.); Wuniri@bjfu.edu.cn (N.W.)
2   Center for East Asian-Australasian Flyway Studies, Beijing Forestry University, Beijing 100083, China
3   Construction and Administration Bureau of South-to-North Water Diversion Middle Route Project,
    Beijing 100038, China; renbingshu@nsbd.cn
*   Correspondence: jiayifei@bjfu.edu.cn; Tel.: +86-010-62336935
†   These authors contributed equally to this work.

**Abstract:** As natural wetlands are degrading worldwide, artificial wetlands can operate as a substitute to provide waterbirds with refuge, but they cannot replace natural wetlands. Reservoirs, one of the most common artificial wetlands in China, can be of great importance to waterbirds. Miyun reservoir in Beijing, China, has undergone a process similar to a natural lake being constructed in a reservoir. In this study, we surveyed waterbird community composition and evaluated the corresponding land cover and land use change with satellite and digital elevation model images of both before and after the water level change. The results showed that in all modelled scenarios, when the water level rises, agricultural lands suffer the greatest loss, with wetlands and forests following. The water level rise also caused a decrease in shallow water areas and a decline in the number and diversity of waterbird communities, as the components shifted from a shallow-water preferring group (waders, geese and dabbling ducks) to a deep-water preferring group (most diving ducks, gulls and terns). Miyun reservoir ceased to be an important waterbird habitat in China and is no longer an important stopover site for white-naped cranes. A similar process is likely to occur when a natural lake is constructed in a reservoir. Therefore, we suggest that policymakers consider the needs of waterbirds when constructing or managing reservoirs.

**Keywords:** Miyun reservoir; waterbird community; hydrology; LCLU; reservoir

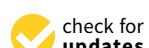



## 1. Introduction

Natural wetlands have been declining rapidly worldwide since the 20th century [1]. In China, extensive loss of natural wetlands has taken place nation-wide since the late 1950s, among which riverine and lacustrine wetlands suffered the greatest loss [2]. In contrast, artificial wetlands, mostly pools and reservoirs, have been increasing significantly [2]. Studies have shown that artificial wetlands can provide waterbirds with substitutes for natural wetlands and shelter in years with food shortage [3–5]. For example, agricultural wetlands, such as lotus ponds and paddy fields, provide refuge for cranes [6]; aquacultural ponds and artificial salt pans provide high-tide roosting sites for costal shorebirds [7,8], and; geese utilize artificial wetlands when drought occurs [9].

Reservoirs and ponds comprise a large proportion of the artificial wetlands in China [2], and to some degree, can complement the loss of natural wetlands for some migratory waterbirds along the East-Asian Australasian Flyway (EAAF) by providing large and stable water bodies [3,10]. Artificially controlled hydrological fluctuations in reservoirs change the habitat and affect the waterbird distribution [11–14]. In a natural environment, waders, such as cranes, egret, herons and shorebirds that utilize the littoral zone of a lake

or reservoir, require large areas of shallow waters and struggle in sites with a frequently fluctuating water level [15,16]. Geese and dabbling ducks prefer a stable water level and unflooded grassland [17–19]. Diving birds require large areas of deep water to forage [10]. Therefore, different hydrological management in reservoirs may benefit different groups of waterbirds.

Functionally, ponds and reservoirs provide supplementary habitats to waterbirds, however, they cannot replace natural wetlands [20]. These artificial wetlands have a lower capacity for waterbirds than natural ones [21]. Waterbirds mainly utilize artificial wetlands when natural habitats fail to meet their survival needs [6,17]. Therefore, converting a natural lake into reservoirs can have a great impact on the waterbird community. Normally, the conversion to reservoirs for water storage results in a rapid water level rise and the large loss of littoral habitats. Studies have shown that rising water levels may benefit a few waterbird species, especially those who utilize deep water bodies, such as grebes and diving ducks [10,12,22]. However, most waterbirds prefer relatively shallow water (geese, herons, bitterns, cranes, swans, and shorebirds) [10,13]. For the latter, a rising water level can cause loss in foraging and resting grounds, such as mudflats, sandbanks, gravel bars and small islands, and grasslands [23]. The water level rise may have caused the natural lake to transform from a far more diversified habitat made of agricultural fields, grasslands, mudflats, sand banks, swamps, shallow water, and deep open water into a uniform habitat that mainly consists of deep open water. Therefore, a reduction in waterbird diversity seems inevitable.

The ecological roles of reservoirs to waterbirds have been well established by previous studies [3,10,12,21,24]. However, the impacts of converting a natural lake to a reservoir with a rapid water level rise on wildlife has only been studied on areas of vegetation and aquatic wildlife [25–30]. Few studies have looked into changes in waterbird communities and their habitats despite the crucial importance of this to policymakers for the conservation of species. One plausible reason for this is the difficulty in obtaining waterbird data before a reservoir is constructed. A study of Pong dam reservoir compared the waterbird status before and after the construction of the reservoir by comparing survey data to old records [31]. The ununified effort of survey between the comparison may result in a biased conclusion. The construction of reservoirs by converting and filling natural lakes can impact the waterbird community. The conversion process can be simplified into a rapid and acute rise of water. Miyun reservoir has been functioning for decades. It has formed stable habitat similar to a natural lake. By studying the process of rapid water level change in Miyun reservoir, we aim to demonstrate how the conversion from a natural lake to a reservoir affects the waterbird community. We aim to investigate how water level changes affect waterbird habitats and communities, in order to provide a better insight into how the construction of a new reservoir affects the waterbird community and which waterbird needs should be taken into consideration in future management.

## 2. Materials and Methods

In this study, we explored habitat change by analyzing the land cover/land use (LCLU) change using satellite images and a geographic information system (GIS). We studied how the waterbird community responded to the above changes by conducting a waterbird survey and analyzing the composition of water depth preference in their community.

We divided our study into two periods for clarification. The two study periods separately represent the status of the Miyun Reservoir before and after the water level rise. The first period, of 2011 to 2014, representing the "before" status of the Miyun Reservoir, and the second "after" period starts in 2018 and ends in 2021.

### 2.1. Study Area

With a total capacity of 4.375 billion m$^3$ [29], the Miyun Reservoir is the largest in the Beijing-Tianjin-Hebei region, north China. It is located in the middle of the Miyun District, north Beijing (40°29′ N, 116°49′ E). With a lack of natural lakes to support waterbirds in the

semi-arid Beijing-Tianjin-Hebei region (Jing-Jin-Ji), the Miyun Reservoir is of great importance to waterbird communities as a substitute, especially for the migratory waterbirds in the EAAF.

Miyun reservoir has been in use for 63 years since its construction in 1958. In the first five decades to 2014, when it received water from South-to-North Water Transfer Schemes (STNWTS), the Miyun reservoir has established stable waterbird habitats, including lowland croplands and forests surrounding the reservoir, large areas of a littoral zone, and a shallow water zone, serving ecological functions similar to a natural lake. It was a biodiversity hotspot, as well as an important stopover site for migratory birds along the EAAF. For example, the white-naped crane (*Antigone vipio*), a vulnerable species according to International Union for Conservation of Nature's Red List (IUCN red list), utilizes the Miyun Reservoir as a stopover site [32,33]. During a survey in 2013, 1020 white-naped cranes were recorded in the Miyun Reservoir [34]. According to the IUCN red list, a mere 500 to 1000 individuals of this species were estimated to winter in China, indicating that almost all members of the population use the Miyun Reservoir as a stopover site.

In December 2014, the Miyun Reservoir, which is the main temporary storage of the middle route of the STNWTS, received a huge quantity of water [35]. According to the Beijing Water Authority, its water level rose approximately from 135 m to 155 m (Figures 1 and 2) [36]. Before the water level rise, the highest water level was 138.13 m, while after the STNWTS period, the highest was 155.30 m. The water level change between the two study periods was approximately 17 m, which is much higher than the average annual water level fluctuation of 3.43 m during the "before" period.

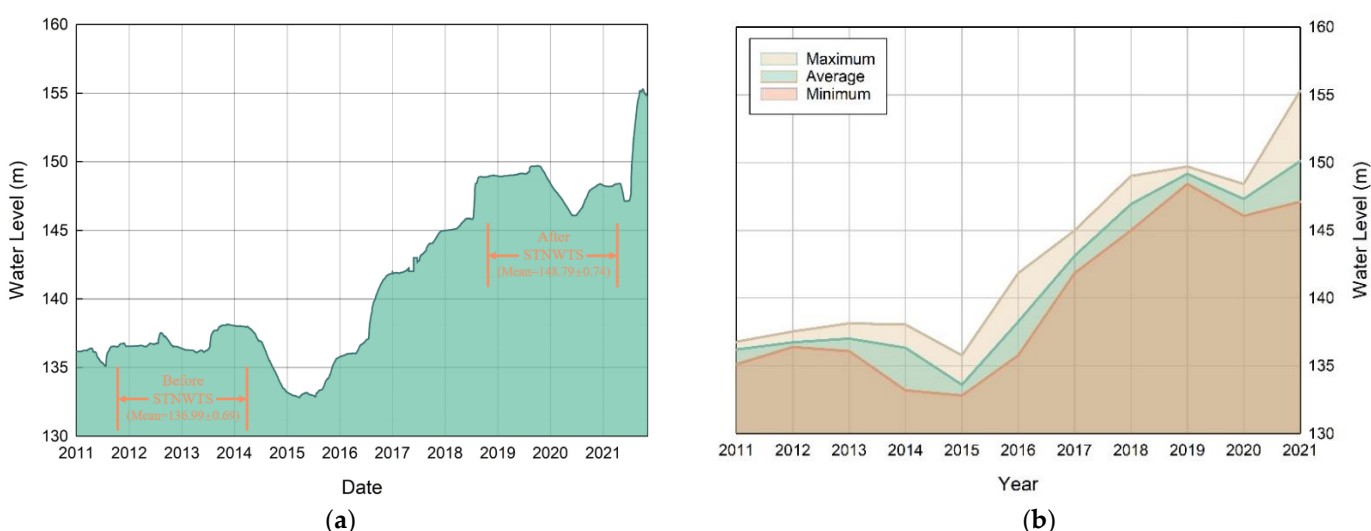

**Figure 1.** (**a**) Daily water level of the Miyun Reservoir from 2011 to 2021. The orange color marks the two study periods of this study, with the mean water level and standard deviation during the two periods. (**b**) The annual minimum, maximum and average water level of the Miyun Reservoir from 2011 to 2021. Data were obtained from Beijing Water Authority official website (http://nsbd.swj.beijing.gov.cn/dzxsksq.html (accessed on 5 November 2021)).

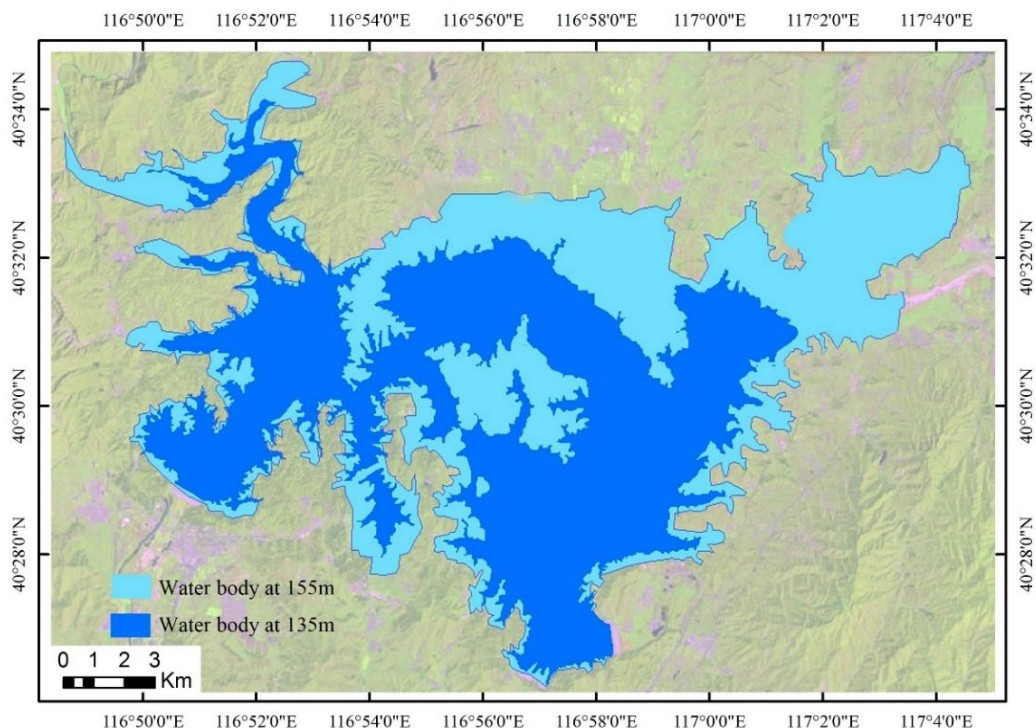

**Figure 2.** The growth of the water body after the SNWTS.

Since its construction, the Miyun Reservoir had never experienced such a dramatic and acute water level change. The rapid water level rise of the Miyun Reservoir can be considered as a newly built reservoir being filled. The water diversion project provided us with a great opportunity to study a case scenario very similar to that of constructing a new reservoir and provide a better understanding of the ecological influences of converting a natural wetland into a manmade reservoir.

### 2.2. Waterbirds Survey

During the two periods, waterbird surveys were conducted at least once per month during the migratory seasons (Table 1).

**Table 1.** The surveying months during each year of the two study periods.

|  | Before STNWTS | | | | After STNWTS | | | |
|---|---|---|---|---|---|---|---|---|
|  | **2011** | **2012** | **2013** | **2014** | **2018** | **2019** | **2020** | **2021** |
| Jan |  |  | √ | √ |  | √ | √ |  |
| Feb |  | √ | √ |  |  | √ | √ |  |
| Mar |  | √ | √ | √ |  | √ | √ | √ |
| Apr |  | √ | √ | √ |  | √ | √ | √ |
| May |  | √ | √ | √ |  | √ | √ | √ |
| Jun |  |  | √ |  |  | √ | √ |  |
| Jul |  |  |  |  |  | √ | √ |  |
| Aug |  |  |  |  |  |  | √ |  |
| Sep |  | √ | √ |  |  | √ | √ |  |
| Oct | √ | √ | √ | √ | √ | √ | √ |  |
| Nov | √ | √ | √ |  | √ | √ | √ |  |
| Dec |  | √ | √ |  | √ | √ | √ |  |

A "√" represents that there was at least one survey conducted during the corresponding month in the corresponding year.

The survey was conducted only on the north shore of the Miyun Reservoir. However, the area is well known for its rich vegetation, abundance of potential food for waterbirds,

and high biodiversity at the beginning of the study [34,37,38]. Therefore, the north shore functions as a good representation of the overall status of the Miyun Reservoir.

The surveys were conducted along three transects which were selected because of their high biodiversity and good representation of the northern bank (Figure 3). During the surveys, Lava 10 × 42 Pro binoculars and Leica APO-Televied 82 telescopes were used for observation. Waterbirds were identified using *A Field Guide to the Birds of China* [39]. We recorded the species and number of individuals, visibility, and human disturbances.

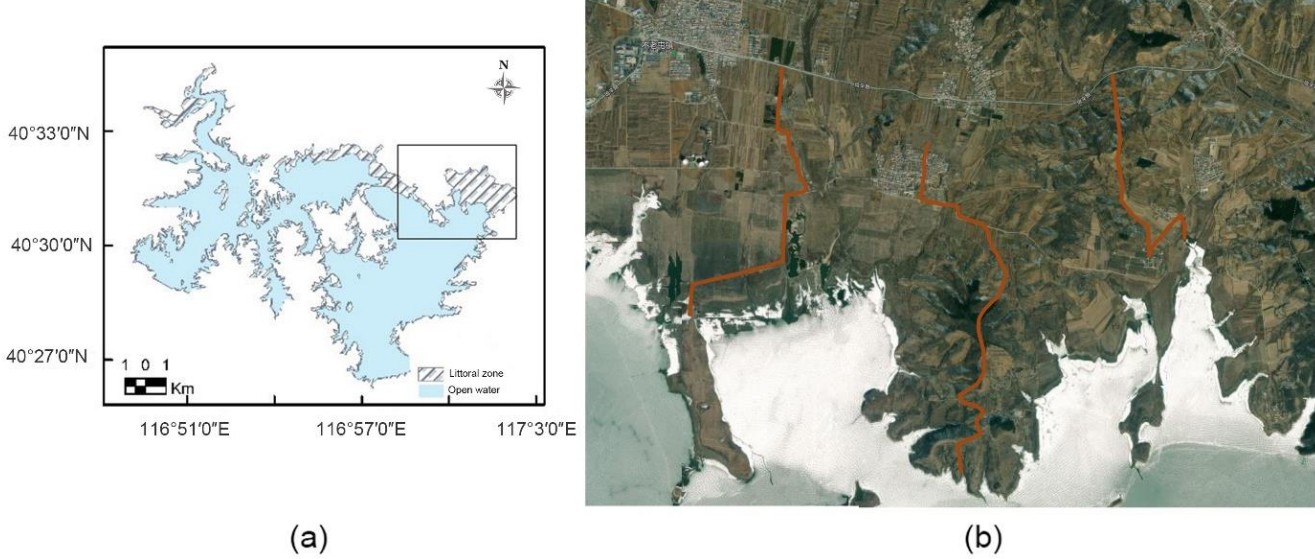

**Figure 3.** Survey area: (**a**) location of the Miyun Reservoir; (**b**) survey area and transect.

### 2.3. Waterbird Analysis

The data collected were analyzed for the α diversity in the spring and autumn migratory season of each year. Four diversity indices were calculated and analyzed to show the change of the α diversity in waterbird community before and after STNWTS, including Simpson's diversity index, Shannon-Wiener diversity index, Marglef's richness index, and Pielou's evenness index.

Waterbirds were categorized into three groups: the shallow-water preferring group, the moderate-depth-water preferring group, and the deep-water preferring group (Appendix A). The highest count of each species during each migratory season is used to calculate the proportion of each water-depth preferring group. Mean values are then calculated among the spring migratory season each year before the water level rise, and after. The same calculations are made for autumn.

### 2.4. Land Cover and Land Use

In this study, Thematic Mapper (TM) images and digital elevation models (DEM) were used to simulate and analyze the inundation process of the Miyun Reservoir. Both the TM images and DEM data have a 30 m resolution. Based on the water level records of the Miyun Reservoir for eleven years (2011–2017), the TM image on 2 May 2015 was selected to analyze the LCLU at the lowest water level (133.12 m) under the natural state, representing the maximum wetland area. The TM image of 21 August 2012 was selected to classify the LCLU at the highest water level under the natural conditions (137.44 m), representing the minimum wetland area. Note that open water was excluded from wetland area calculation. For the purpose of the study, the word "wetland" that we adopted for the LCLU analysis is slightly different from the Ramsar definition [40]. Here it is mainly comprised of littoral zones, which are the areas that are seasonally inundated with water and are the transitional zones between terrestrial and aquatic environments. In the Miyun Reservoir, this includes

mudflats, sandbanks and waterside swamps. ERDAS IMAGE and ArcGIS10.0 were used for supervised classification and visual interpretation of the two images, and the LCLU map of the study area was obtained. On-site observations during both the highest and lowest water levels, under natural conditions, were also made to ensure the accuracy of the classification and interpretation of the two images. As the highest water level reached was 155.3 m during our study period, the water levels of 145 m, 150 m, and 155 m were selected to simulate the inundation process of the Miyun Reservoir based on the 1:10,000 contour map of the Miyun Reservoir. The inundation area of each type of landcover during the process of the water level rising was calculated with ArcGIS10.0.

## 3. Results

### 3.1. Hydrology Driven Habitat Change

As shown in Table 2, based on the water level on 24 October 2012, when the water level reached 145 m, the wetland decreased by 161.96 ha, the forest area decreased by 162.30 ha, and the agricultural land area decreased by 4247.13 ha. When the submerged water level reached 150 m, the wetland decreased by 161.96 ha, the forest area decreased by 314.68 ha, the agricultural land area decreased by 5826.37 ha, and the built-up land decreased by 0.21 ha. When the submerged water level reached 155 m, the wetland decreased by 161.96 ha, the forest area decreased by 538.45 ha, the agricultural land area decreased by 6971.39 ha, and the built-up land decreased by 3.66 ha.

**Table 2.** Areas flooded during each scenario.

| Simulation Scenario | Land-Use Type | 2012 (137.44 m) | 2015 (133.12 m) |
|---|---|---|---|
| Area flooded when water level rise to 145 m | Wetland | 161.96 (3.54%) | 1083.72 (18.08%) |
| | Forest | 162.30 (3.55%) | 235.64 (3.93%) |
| | Agricultural | 4247.13 (92.91%) | 4675.12 (77.99%) |
| Area flooded when water level rise to 150 m | Wetland | 161.96 (2.57%) | 2167.44 (18.08%) |
| | Forest | 314.68 (4.99%) | 471.28 (3.93%) |
| | Agricultural | 5826.37 (92.43%) | 9350.24 (77.99%) |
| | Built-up | 0.21 (<0.01%) | |
| Area flooded when water level rise to 155 m | Wetland | 161.96 (2.11%) | 2167.49 (16.42%) |
| | Forest | 538.45 (7.02%) | 719.40 (5.39%) |
| | Agricultural | 6971.39 (90.83%) | 10,459.42 (78.37%) |
| | Built-up | 3.66 (0.05%) | 3.45 (<0.01%) |

Based on the water level on 2 May 2015, when the submerged water level reached 145 m, the wetland decreased by 1083.72 ha, the forest area decreased by 235.64 ha, and the agricultural land area decreased by 4675.12 ha. When the submerged water level reached 150 m, the wetland decreased by 2167.44 ha, the forest area decreased by 471.28 ha, and the agricultural land area decreased by 9350.24 ha. When the submerged water level reached 155 m, the wetland decreased by 2167.49 ha, the forest area decreased by 719.40 ha, the agricultural land area decreased by 10,459.42 ha, and the built-up land decreased by 3.45 ha.

### 3.2. Change in Waterbird Community Structure

We found a decline in the $\alpha$ diversity of the waterbird community after the water level change, and a transition in community structure.

The maximum and average $\alpha$ biodiversity indexes are lower after the STNWTS period compared to before, except for the average value for Margalef's richness index (Table 3). With a maximum of 0.15 and an average of 0.02, Margalef's richness index varies greatly between surveys in the period before STNWTS.

**Table 3.** Biodiversity indexes of waterbirds during the two study periods.

| | Maximum | | Average | | *p*-Value |
|---|---|---|---|---|---|
| | Before STNWTS | After STNWTS | Before STNWTS | After STNWTS | |
| Shannon-Wiener Diversity Index | 3.10 | 2.55 | 1.79 | 1.42 | 0.07 |
| Simpson's Diversity Index | 0.93 | 0.89 | 0.73 | 0.58 | 0.15 |
| Margalef's Richness Index | 0.15 | 0.09 | 0.02 | 0.03 | <0.01 ** |
| Pielou's Evenness Index | 0.84 | 0.82 | 0.58 | 0.55 | 0.83 |

** *p* value < 0.01

By collecting information on the threshold water depth (the deepest water level that a waterbird species can endure or utilize) for each waterbird species, based on information from the IUCN Redlist, *Fauna of China*, and previous studies [10,22,41–55], we arbitrarily divided the threshold depth into three groups. The species were divided into different water level preferring groups according to their threshold depth. The species with a threshold depth of 10 m or deeper were divided into the deep-water preferring group, while the species with a threshold depth between 0.9 m to 7 m were divided into the moderate-depth preferring group, and those with a threshold depth below 0.6 m were divided into the shallow-water preferring group (Figure 4).

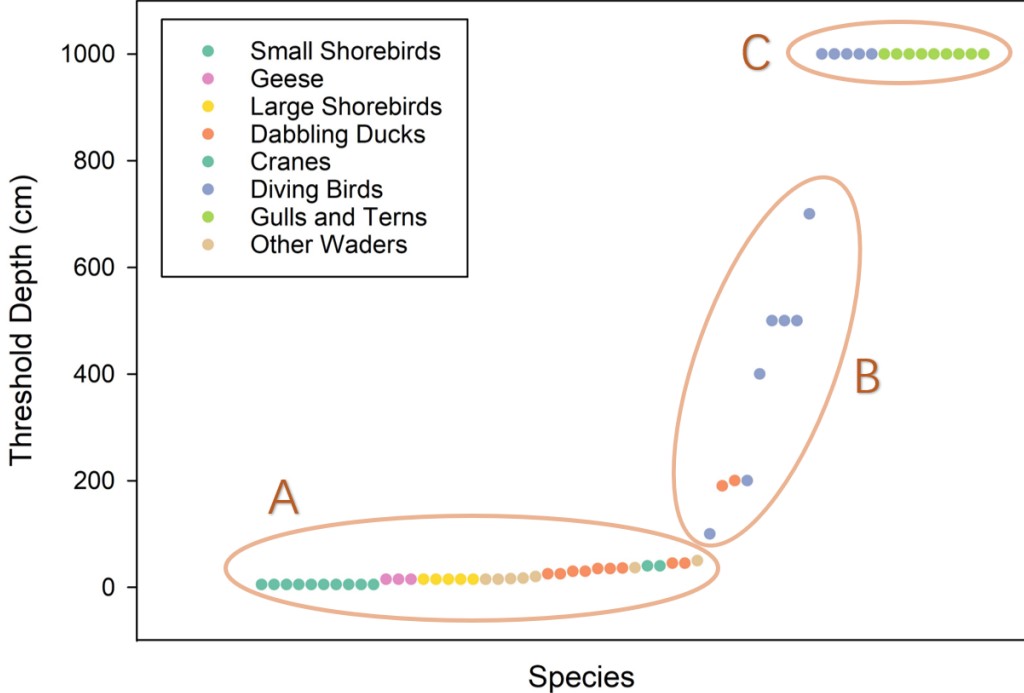

**Figure 4.** Threshold depths of each species. The threshold naturally distributed to three groups. Water level preference groups were categorized accordingly. A, B, and C sequentially represent the shallow-water preferring group, the moderate-depth-water preferring group, and the deep-water preferring group.

In our three water depth preferring groups, the shallow-water preferring group mainly consisted of waders, geese, and most dabbling ducks. The moderate-depth-water preferring group consists of other dabbling ducks, Whooper Swan (*Cygnus cygnus*) and a few diving birds. The deep-water preferring group consists of the remaining diving birds, gulls, and terns. A complete list of the species recorded during all surveys and the water level preference for the recorded species of the selected dates can be found in Appendix A.

The waterbird community showed an overall preference for shallow water before the STNWTS period, while afterwards, it showed an overall preference for deep water. Before the STNWTS period, the shallow preferring group was the most dominant (Figure 5). However, after the STNWTS period, there was a tendency for a less shallow water preferring group, and more individuals preferred deeper water. As such, the Miyun Reservoir has lost a great proportion of waterbirds that prefer shallow water.

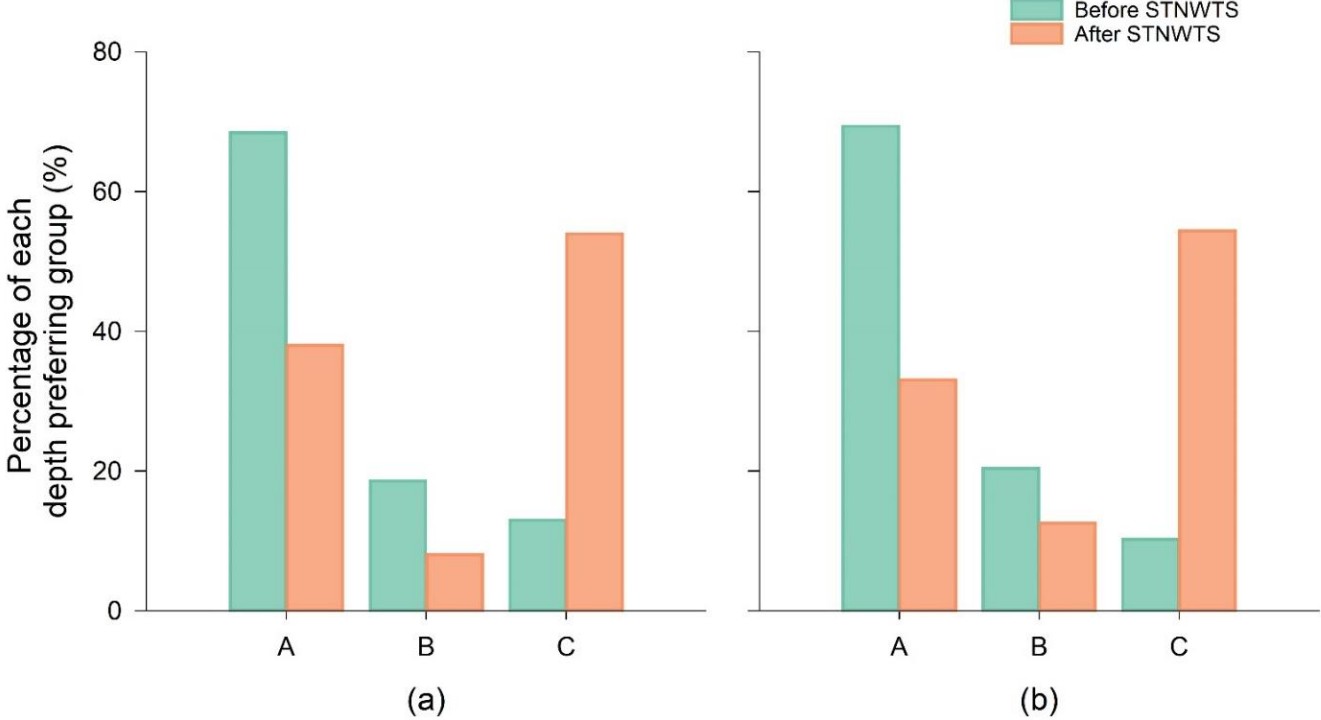

**Figure 5.** The percentage of each depth preferring group during the two study periods: (**a**) spring migratory season; (**b**) autumn migratory season. A: shallow-water preferring group, with threshold depths all below 0.6 m; B: moderate-depth-water preferring group, with threshold depths between 0.9–7 m; C: deep-water preferring group, with threshold depths over 10 m.

### 3.3. Change in Waterbird Population

We have found a decline in the species variety and population size of waterbirds after the water level change. In the period before STNWTS, the highest waterbird count was 10,436 individuals. On average, 2981.39 individuals were recorded during each survey. In the period after STNWTS, the highest waterbird count was 1538 individuals, with an average of 578.41 individuals for each survey. Our findings indicate a more than 80% decline in the population between the two study periods.

Among these, a few IUCN threatened species have ceased to inhabit the reservoir. During the period before STNWTS, 105 waterbird species were recorded, including two critically endangered species (CR), three endangered species (EN), eight vulnerable (VU) species and eight species exceeding the 1% population criteria. During the period after STNWTS, 58 waterbird species were recorded. Of the 58 species, only three vulnerable species were recorded and only two species exceeded the 1% population criteria. All of the threatened cranes and geese ceased to inhabit the reservoir. The critically endangered Baer's pochard (*Aythya baeri*), vulnerable relic gull (*Larus relictus*) and horned grebe (*Podiceps auritus*) were also not recorded in the period after STNWTS. Three vulnerable species continued to inhabit the reservoir, including the common pochard (*Aythya ferina*), swan goose (*Anser cygnoid*) and velvet scooter (*Melanitta fuscua*). The two former species suffered a drop in population after the water level rise, while the latter has only one individual recorded in each study period. The species with a population over the 1% criteria either

ceased to inhabit the reservoir, or dropped below 1%. In contrast, the number of great crested grebe (*Podiceps cristatus*) and black stork (*Ciconia nigra*) increased in the period after STNWTS, exceeding the 1% criteria.

## 4. Discussion

### 4.1. The Classification of Baer's Pochard

Previous studies showed that Baer's pochards *(Aythya baeri)* are only active in water depths of less than 2 m [52]; however, the species is still categorized as a deep-water preferring group, and we consider that the low water depth threshold is likely to be the result of inadequate observation. It is generally difficult to observe the activities of pochards below 4 m, yet a previous study confirmed that almost all pochards are active in deep waters [53]. However, Baer's pochard category does not interfere with our study's conclusion, as the analysis shows the same trend, in that Baer's pochard can be either the shallow-water preferring group or deep-water preferring group.

### 4.2. Hydrology Driven Habitat Degradation

Water level determines the type and availability of habitats for waterbirds [56–58]. Water level determines the availability of areas with different water depths, which greatly effects the distribution of waterbirds [59]. Dabbling waterfowls and waders need shallow water areas to feed [60], whereas diving birds require a minimum depth to perform diving [10]. Water levels can also indirectly affect habitat availability by impacting food resources (vegetation, fish, etc.) and shelter vegetation for waterbirds [61,62]. Water level fluctuation can create littoral zones which are preferred by waders, although frequent fluctuation is disliked by diving birds [10].

In the process of the water level rise in the Miyun reservoir, the habitat went through a series of changes, including the loss of both suitable LCLU and shallow water areas. The Miyun Reservoir is an isolated habitat surrounded by urban and built-up land. As shown in the result of our LCLU analysis, when the water level rose in the Miyun Reservoir, 45,713,891 m$^2$ of land were flooded, including large areas of littoral zones, croplands and forests, decreasing the areas of suitable habitat. Our land cover and land use analysis did not provide results on the change of habitat with preferable water depth, because DEM products usually see areas of water bodies as a flat surface with the same evaluations [63], and cannot distinguish between shallow and deep water. Our results regarding the waterbird water depth preferences indicate that the shallow water area has decreased. We found that in the period before STNWTS, the waterbirds showed an overall tendency towards shallow water, while in the period after STNWTS, they showed an overall tendency towards deep water. This indirectly proves that the shallow water area has decreased. In conclusion, the littoral zone, agricultural land, forests and shallow water habitat has decreased as a result of the rising water level.

### 4.3. Decline in Waterbird Diversity and Shift in Community Structure as a Response

Waterbirds are strongly affected by the above habitat changes. Our study showed a decline in diversity and the transition of water level preferences in a water bird community after the water level change.

Our results overall for LCLU show that agricultural lands suffered the most loss. Agricultural lands have proven to be a great habitat for cranes [6], herons and egrets [5], and geese [64]. According to our observation, the agricultural land lost due to the rise in water level is mostly corn and alfalfa fields. During the harvest season, these lands are flat and have few obstacles. The residual corn has been established as an important food resource for many waders and geese [64,65]. The loss of agricultural land explains the steep drop in the number of cranes and geese during the period after STNWTS.

The loss of littoral zones, though not occurring as much as the loss of agricultural and forest land, might be more severe for these animals. The loss of littoral zones indicates a loss of mudflats, sandbanks, and waterside swamps. It is well established that many

waterbirds forage in these habitats [3,10,16,22,51]. The loss of littoral zones may indicate a direct loss of dietary resources, reducing the capability of the Miyun Reservoir to serve as an important stopover site for migratory and/or wintering waterbirds.

The loss of shallow water areas changed the overall water level preference in the waterbird community. In most regions around the world, waterbird communities mostly consist of waterbirds which prefer shallow water [10,66]. The waterbird community in Miyun also conformed to this pattern: in all waterbirds, the shallow water preferring group with a threshold depth of up to 0.5 m consisted of the highest proportion of the community. In the comparison of the two study periods, the community composition of the Miyun Reservoir transformed from more individuals from the shallow water preferring group to more from the deep-water preferring group. It can be inferred that this transformation is caused by the loss of the shallow water habitat when the water level rises.

### 4.4. Miyun Reservoir Ceased to be an Important Waterbird Site in North China

The degradation of habitat reduced the importance of the Miyun Reservoir to waterbirds. Before STNWTS, the Miyun Reservoir was one of the most important waterbird habitats. Its importance is similar to that of Yeya Lake and Bei Dagang, where the former is the only natural reserve of wetland, and the latter is a wetland of international importance. Before STNWTS, the overall number of species, and the proportion of IUCN threatened species and nationally protected species, were between the numbers found in Yeya Lake and Bei Dagang [67,68] (Table 4). In 2014, which is the year before the Miyun Reservoir started receiving water, a single waterbird count was as high as 22,617 individuals, with 10 species exceeding their 1% population criteria, reaching the standard of wetlands of international importance. Thus, the Miyun Reservoir was significantly important for waterbirds. After the water level rose, the number of species, the overall abundance of waterbirds, and biodiversity decreased. Since then, there has not been a single waterbird count with individuals over 10,000, and only two species exceeded their 1% criteria. These observations indicate that the Miyun Reservoir ceased to be an important waterbird habitat in north China and can no longer match Yeya Lake and Bei Dagang.

**Table 4.** The total species count, the percentage of IUCN endangered species, the percentage of national protected animals before and after STNWTS, and their comparison to Beijing's Yeya Lake and Tianjin's Bei DaGang wetland.

| | Yeya Lake [1] | Bei Dagang [2] | Miyun Reservoir Before STNWTS | Miyun Reservoir After STNWTS |
|---|---|---|---|---|
| Overall number of species | 103 | 143 | 105 | 58 |
| IUCN threatened species | 9.62% | 13.89% | 12.38% | 5.17% |

[1] The data for Beijing's Wild Duck Lake is cited from a survey led by QU Yuanyuan during 2004–2009. [2] The data for Tianjin's Bei DaGang Wetland is cited from a survey led by CHAI Ziwen during 2017–2019.

### 4.5. Miyun Reservoir is No Longer an Important Stopover Site for White-Naped Cranes

The degradation of habitat also impacted on the use of the reservoir as an important stopover site for the white-naped crane. The Miyun Reservoir played an important ecological role in the Beijing-Tianjin-Hebei region because it was an important stopover site for white-naped cranes and many other shorebirds among its range [33]. In 2013, we found that almost the entire west population of white-napped cranes used the Miyun Reservoir as a stopover, which was a unique phenomenon in Beijing's wetlands [34]. After the water level change, we did not find white-naped cranes in the Miyun Reservoir, and found only a handful of other waders. The Miyun Reservoir may no longer be a suitable site for white-naped cranes. If the above is true, white-naped cranes have lost important feeding and refueling grounds. In recent years, Beijing has not recorded white-naped cranes with populations of over 1000. A recent study using satellite tracking found that white-naped cranes used the upper reaches of the Luanhe River north of Miyun Reservoir

as their stopover site [69]. This might be one of the consequences of hydrological changes in the Miyun Reservoir.

### 4.6. Study Results Give Insight into the Process of Constructing a New Reservoir

The Miyun Reservoir has been in functional use for 58 years, making its ecology features similar to that of a natural lake. The rapid water level rise due to STNWTS is similar to that of a natural lake modified into a reservoir filled with water. Therefore, our study provides insight into the construction of a new reservoir with its previous natural status. It is possible that the changes that happened in the Miyun Reservoir are going to repeat themselves on newly built reservoirs. Artificial wetlands have been proven to serve as important substitutes for natural wetlands [4,6,10,31], yet they cannot completely replace natural wetlands [20,21]. According to our study, constructing a natural lake in a reservoir may cause degradation in habitat and have a mostly negative impact on its waterbird community, which may endanger its ability to provide ecosystem services. These are the risks that policymakers should consider when establishing a new reservoir.

Few studies have been conducted on the effect of reservoir construction on waterbirds. However, our results are in conflict with a previous study on the construction of the Pong dam reservoir, India [31]. This study compared the waterbird status before and after the construction of the reservoir by comparing survey data to old records, and reached the conclusion that the construction of the Pong dam reservoir benefits waterbirds due to the extensive drainage of natural wetlands in its region. There are several possible reasons for this conflict. First, old records may be unreliable. Second, the bird survey was conducted about 20 years after the reservoir's construction. With good wetland management, the reservoir can recover to a similar state to a natural lake. Last, the extensive degradation of natural wetlands in the region makes the case of the Pong dam reservoir unique and unrepresentative.

Further research is needed on the vegetation change in the periods before and after STNWTS, in order to better understand the mechanism of waterbird responses.

### 4.7. Recommendations for Reservoir Management

Reservoirs and their water regulations are double edged swords for waterbirds [3]. While converting a natural lake into a reservoir can lead to biodiversity loss and mask the loss of natural wetlands [70], insightful structures and good management can allow reservoirs to provide waterbirds with stable open water. In the world of rapid modernization, it is important to find the balance between exploiting and managing natural resources according to human needs and nature reservation. The Pong dam reservoir in India is an adequate example of a waterbird-friendly reservoir [31,71]. It was constructed in 1985 with several flat and dry banks, with little vegetation and shallow water areas. It has maintained significant biodiversity in India and shelters several hundred waterbird species. Due to its importance to waterbirds, it has been declared a bird sanctuary. This evidence proves that reservoir management does not necessarily conflict with waterbird conservation and can provide shelter for waterbirds who have fewer habitats available.

Before STNWTS, the Miyun Reservoir was also in a state similar to the Pong dam reservoir. It was one of the locations with the highest waterbird diversity in Beijing, and it was an important refueling site for many migratory birds. Therefore, the Miyun Reservoir used to be subject to good artificial wetland management. However, regulating water without acknowledging the needs of waterbirds can cause serious habitat loss for them. In the case of the Miyun Reservoir, species that prefer shallow water are the first in line to suffer the consequences. In the engineering of the South-to-North Water Diversion Project, the effect of water level changes on waterbirds was overlooked, thus causing a major deterioration in waterbird habitat and a decline in the importance artificial wetlands as waterbird habitat.

In the management of an artificial wetland water level regime, it has been suggested that a combination of shallow water areas and deep water should be provided, and the

water level should be closely monitored and controlled during the migratory season to accommodate waterbird needs [60,61]. In those studies, a need for shallow water areas was highlighted. Therefore, to ensure the water storage requirements of the Miyun Reservoir, we suggest that targeted habitat restoration should be conducted, such as terrain reconstruction, slope mitigation, or construction of ecological islands to provide possible shallow water areas. Transforming artificial wood back into natural grasslands and flats on the bank is also a necessity for geese and waders.

Overall, while managing a reservoir, the impact of water level change on waterbird communities should always be taken into account. A higher waterbird diversity requires a diverse habitat, among which grasslands and shallow water are the most important. The bank should have a gentle slope, providing as much shallow water below 3 m as possible. A few chosen areas should be planted with emergent plants to provide breeding grounds and shelter for waterbirds in need.

## 5. Conclusions

Our study showed that the water level change in the Miyun Reservoir caused the number of waterbird species and individuals to drop rapidly, and the constitution of the waterbird community shifted from a shallow water preferring group to a deep-water preferring group. We also observed a decline in importance of the Miyun Reservoir as a waterbird habitat. We believe that a similar process is likely to occur when a natural lake is modified into a reservoir. Therefore, we suggest that stakeholders take the needs of waterbirds into account when considering the construction of new reservoirs to avoid the loss of waterbird diversity. Future management of reservoirs after their construction should also take into account the needs of a waterbird habitat to create an artificial wetland with high supporting services.

**Author Contributions:** Conceptualization, W.L., J.L. and Y.J.; methodology, Y.J.; validation, W.L., J.L., B.R., R.C., Z.Y. and N.W.; writing, W.L. and Y.J.; supervision, Y.J. All authors have read and agreed to the published version of the manuscript.

**Funding:** This research was funded by National Natural Science Foundation of China (No. 31971400), the first class General Financial Grant from the China Postdoctoral Science Foundation (No.2017M620023) and the Beijing Forestry University "National Training Program of Innovation and Entrepreneurship for Undergraduates" (No. 202010022160). We wish to thank the Free Flying Wings Program, and the SEE Foundation.

**Institutional Review Board Statement:** Not applicable.

**Informed Consent Statement:** Not applicable.

**Data Availability Statement:** Please refer to suggested Data Availability Statements in section "MDPI Research Data Policies" at https://www.mdpi.com/ethics, accessed on 5 November 2021.

**Conflicts of Interest:** The authors declare no conflict of interest.

## Appendix A

**Table A1.** Species recorded in all surveys and water depth preferences during the four pairs of selected dates. A, B, C sequentially represents the shallow-water preferring group, the moderate-depth-water preferring group, and the deep-water preferring group.

| Common Name | Scientific Name | Water Depth Preference Group |
| --- | --- | --- |
| Baer's Pochard | *Aythya baeri* | C |
| Baikal Teal | *Sibirionetta formosa* | A |
| Bar-headed Goose | *Anser indicus* | A |
| Bar-tailed Godwit | *Limosa lapponica* | A |
| Bean Goose | *Anser fabalis* | A |
| Black Stork | *Ciconia nigra* | A |
| Black-crowned Night Heron | *Nycticorax nycticorax* | A |
| Black-headed Gull | *Larus ridibundus* | C |
| Black-necked Grebe | *Podiceps nigricollis* | B |
| Black-tailed Godwit | *Limosa limosa* | A |
| Black-tailed Gull | *Larus crassirostris* | C |
| Black-winged Stilt | *Himantopus himantopus* | A |
| Brown-headed Gull | *Larus brunnicephalus* | C |
| Caspian Tern | *Hydroprogne caspia* | C |
| Cattle Egret | *Bubulcus ibis* | A |
| Chinese Pond Heron | *Ardeola bacchus* | A |
| Common Coot | *Fulica atra* | B |
| Common Crane | *Grus grus* | A |
| Common Goldeneye | *Bucephala clangula* | C |
| Common Greenshank | *Tringa nebularia* | A |
| Common Merganser | *Mergus merganser* | C |
| Common Moorhen | *Gallinula chloropus* | B |
| Common Pochard | *Aythya ferina* | C |
| Common Redshank | *Tringa totanus* | A |
| Common Sandpiper | *Actitis hypoleucos* | A |
| Common Shelduck | *Tadorna tadorna* | A |
| Common Snipe | *Gallinago gallinago* | A |
| Common Tern | *Sterna hirundo* | C |
| Curlew Sandpiper | *Calidris ferruginea* | A |
| Demoiselle Crane | *Grus virgo* | A |
| Dunlin | *Calidris alpina* | A |
| Eastern Curlew | *Numenius madagascariensis* | A |
| Eastern Spot-billed Duck | *Anas zonorhyncha* | B |
| Eurasian Bittern | *Botaurus stellaris* | A |
| Eurasian Curlew | *Numenius arquata* | A |
| Eurasian Spoonbill | *Platalea leucorodia* | A |
| Eurasian Wigeon | *Mareca penelope* | A |
| Eurasian Woodcock | *Scolopax rusticola* | A |
| Falcated Duck | *Mareca falcata* | A |
| Ferruginous Duck | *Aythya nyroca* | C |
| Gadwall | *Mareca strepera* | A |
| Garganey | *Spatula querquedula* | A |
| Graylag Goose | *Anser anser* | A |
| Great Cormorant | *Phalacrocorax carbo* | C |
| Great Crested Grebe | *Podiceps cristatus* | C |
| Great Egret | *Ardea alba* | A |
| Greater Sand Plover | *Charadrius leschenaultii* | A |
| Greater White-fronted Goose | *Anser albifrons* | A |
| Green Sandpiper | *Tringa ochropus* | A |

**Table A1.** *Cont.*

| Common Name | Scientific Name | Water Depth Preference Group |
| --- | --- | --- |
| Green-backed Heron | *Butorides striata* | A |
| Green-winged Teal | *Anas crecca* | A |
| Grey Heron | *Ardea cinerea* | A |
| Grey Plover | *Pluvialis squatarola* | A |
| Grey-headed Lapwing | *Vanellus cinereus* | A |
| Gull-billed Tern | *Gelochelidon nilotica* | C |
| Hooded Crane | *Grus monacha* | A |
| Horned Grebe | *Podiceps auritus* | C |
| Intermediate Egret | *Ardea intermedia* | A |
| Kentish Plover | *Charadrius alexandrinus* | A |
| Lesser Sand Plover | *Charadrius mongolus* | A |
| Lesser White-fronted Goose | *Anser erythropus* | A |
| Little Egret | *Egretta garzetta* | A |
| Little Grebe | *Tachybaptus ruficollis* | B |
| Little Ringed Plover | *Charadrius dubius* | A |
| Little Tern | *Sternula albifrons* | C |
| Long-billed Plover | *Charadrius placidus* | A |
| Long-toed Stint | *Calidris subminuta* | A |
| Mallard | *Anas platyrhynchos* | A |
| Mandarin Duck | *Aix galericulata* | C |
| Marsh Sandpiper | *Tringa stagnatilis* | A |
| Mew Gull | *Larus canus* | C |
| Northern Lapwing | *Vanellus vanellus* | A |
| Northern Pintail | *Anas acuta* | A |
| Northern Shoveler | *Spatula clypeata* | A |
| Oriental Pratincole | *Glareola maldivarum* | A |
| Oriental White Stork | *Ciconia boyciana* | A |
| Pacific Golden Plover | *Pluvialis fulva* | A |
| Pallas's Gull | *Larus ichthyaetus* | C |
| Pied Avocet | *Recurvirostra avosetta* | A |
| Purple Heron | *Ardea purpurea* | A |
| Red Knot | *Calidris canutus* | A |
| Red-crowned Crane | *Grus japonensis* | A |
| Red-necked Stint | *Calidris ruficollis* | A |
| Relict Gull | *Larus relictus* | C |
| Ruddy Shelduck | *Tadorna ferruginea* | B |
| Ruddy-breasted Crake | *Zapornia fusca* | A |
| Ruff | *Calidris pugnax* | A |
| Sandhill Crane | *Grus canadensis* | A |
| Sharp-tailed Sandpiper | *Calidris acuminata* | A |
| Siberian Crane | *Grus leucogeranus* | A |
| Smew | *Mergellus albellus* | B |
| Spotted Redshank | *Tringa erythropus* | A |
| Swan Goose | *Anser cygnoid* | A |
| Temminck's Stint | *Calidris temminckii* | A |
| Tufted Duck | *Aythya fuligula* | B |
| Tundra Bean Goose | *Anser serrirostris* | A |
| Tundra Swan | *Cygnus columbianus* | A |
| Vegae/Mongolian Gull | | C |
| Velvet Scoter | *Melanitta fusca* | C |
| Water Rail | *Rallus aquaticus* | A |
| Whimbrel | *Numenius phaeopus* | A |
| Whiskered Tern | *Chlidonias hybrida* | C |
| White-naped Crane | *Grus vipio* | A |
| White-winged Tern | *Chlidonias leucopterus* | C |
| Whooper Swan | *Cygnus cygnus* | B |
| Wood Sandpiper | *Tringa glareola* | A |
| Yellow Bittern | *Ixobrychus sinensis* | A |

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
