# Peer review of "The Impacts of a Large Water Transfer Project on a Waterbird Community in the Receiving Dam: A Case Study of Miyun Reservoir, China"

_remotesensing, doi:10.3390/rs14020417_

Round 1

Reviewer 1 Report

This study analyzed the habitat and water bird changes in Miyun Reservoir after an evident water level increase. The topic is important but I recommend the authors to revise the manuscript for a potential publication in the journal Remote Sensing. My concerns are mainly on the following points.

Title, I recommend the authors to improve the current title. I don’t think this study addressed the common issues caused by a large water transfer project. This study only focused on the changes in habitat and water bird number changes caused by the water level increase. This only occurred in the destination of the water transfer project.

Abstract, I really think this section requires the quantified data to support the results and conclusions. Too many backgrounds are present.

Introduction, the authors highlighted the habitat loss due to the conversion from natural wetland to artificial wetland, especially from a small lake to a bigger reservoir. But there is a need to conclude the progress of the related studies on the water birds. Moreover, some texts about the water transfer project and Miyun reservoir should be added.

Data and methods, I don’t think the study area has forested wetland and no marsh. Please give more details on the classification system. How about was the classification accuracy? How did you define the littoral zone of the Miyun reservoir?

Discussion, more discussions should be focused on the relationship of habitat change and water bird change. I don’t think the subtitle of 4.4 is suitable. The 7 subsections should be combined.

Reviewer 2 Report

Review –remote sensing 1506024

The impacts of a large water transfer project on the waterbird community, case study of Miyun Reservoir, China

Major comments

Review comments are divided below into substantive comments followed by editorial comments.

Substantive comments

Under materials and methods- lines 133-134- it would be good for the authors to explain in more detail why the bird surveys were done only at the northeastern shore area.

Under results –line 198 Table 3- the authors should have a brief discussion of why some biodiversity indexes show less or more amount of change, e. g., Pielow’s Eveness index for the earlier period.

Under discussion lines 243-252- It seems like the whole section” The classification of water level preference group” belongs in the previous section –Results?

Lines 352-368 most importantly the authors should discuss or compare study results to any other comparable studies and future research needed.

Editorial comments

Abstract

Line 26- suggest- is likely to occur

Introduction

Line 32- suggest -nation-wide

Line 33- suggest – greatest loss

Line 53- suggest – utilize

Materials and methods

Line 98-suggest – has been in use for

Discussion

Line 294- suggest- Our results overall for LCLU…

Line 299- suggest- the steep drop

Line 307-suggest- mostly consist

Line 352- suggest- 4.6. Study results give insight…

Line 353- suggest- in functional use..

Reviewer 3 Report

This is a straightforward ad most useful study of the changes in a well-studied area close to Beijing. 

While China has protected about 18% of its terrestrial area and “red lines” protect about another 12% to some extent, it has neglected its wetlands. China tends to classify those as unimportant, and they have suffered substantial changes.  Those changes are subtle.  Superficially, China seems to have more “wetland” areas than in the past. Xu et al. show that this misleads: there are more deep reservoirs and fewer natural wetlands.  

Xu W, Fan X, Ma J, Pimm SL, Kong L, Zeng Y, Li X, Xiao Y, Zheng H, Liu J, Wu B. Hidden loss of wetlands in China. Current Biology. 2019 Sep 23;29(18):3065-71.

So what’s the problem with deep reservoirs?  This paper provides a detailed answer.

Miyun lake is close to Beijing, a city that has long enjoyed an active birdwatching community.  The lake is frequently visited and well-known — and has been for over a decade.  Those who manage the reservoir dramatically changed its water levels during that period.  The paper documents those changes well using remote sensing data and land and water cover classifications.  That’s relatively straightforward, and I see no problems with the authors’ methods.  They then relate these changes to those in the well-known bird communities.  These changes harm some globally threatened species. 

The authors’ points are well taken — reservoir management can benefit endangered species if done right.  The study adds to those at one of Asia’s major natural wetlands — Poyang Lake — where managed water flows have harmed other endangered species.

Reviewer 4 Report

The evaluated article entitled "The impacts of a large water transfer project on the waterbird community, case study of Miyun Reservoir, China" concerns a study of changes in the avifauna of a selected water reservoir in China. It is, as the title suggests, a case study and as such provides a valuable example of biodiversity change induced by changes in reservoir water levels. The very problem of how such changes affect organisms, including birds, is relatively well understood and adds nothing new. 
The main reservation concerns the correct addressing of the paper. The authors did not indicate how they used the remote sensing (RS) element in their study. Of course, the methodology mentions the analysis of satellite images and models, but they did not state the quality of these data. After analysing the text, one gets the impression that typical avifauna survey results are presented here and RS is only a minor element. So it is not clear why the authors directed the study to the journal Remote Sensing . This calls for deeper consideration on the part of the authors and definitely a shift of emphasis in the paper. When reading the study, one encounters many ambiguities regarding the authors' overall approach to the issues raised. Much emphasis is placed on hydrological changes (line 79) and their natural consequences. However, it is difficult to find these changes (as a set of factors) apart from changes in water levels. This raises the question of why to use a term that suggests a multifactorial analysis rather than simply describing a change in water levels. The use of the term littoral zone is also not entirely clear. The authors did not specify what the littoral zone is and what depth limit they used to define it. Another issue is the enumeration of many indicators indicating bird diversity. What was the reason for this? Probably it can be rationally explained, but the authors did not do so. 
In the analyses of the designated periods before and after the influx of water, analyses were conducted for different sampling values. Has it been checked how this affects the results? Besides, there are months from which samples were taken only once (July) and 8 times (October). It seems appropriate to select for comparisons only those months where we have a maximum or comparable portion of information. There is no deeper statistical analysis of the empirical data we have, and when tests appear (e.g. Table 3), there is no information about which test was performed. In addition, I have many repetitions of text even in two consecutive paragraphs (e.g. lines 212-216 and 284-289 or 358-360 and 364-365). The manuscript needs a thorough revision. 

Specific comments (most important)
The abstract needs to be revised as it does not present the main achievements of the work.
Line 79 is not clear about hydrological changes 
Line 80 on what basis the authors say they are dealing with a new water body 
Line 93 It is not clear to the international reader where the water body is located
Figure 1 is not clear. How was this graphic created? It is necessary to mark the periods studied and to work out from a hydrological point of view e.g. the average water level.
Figure 2: the legend does not explain anything, neither does the caption (the period - years - should be specified).
Figure 4 Instead of A, B and C the depth should be given or classified. The description of the Y axis is incorrect. Which letter corresponds to which depth range?
Lines 212-216 The passage contains sentences in which the same observation is repeated. Please critically revise the text.
I suggest tabulating information on endangered or valuable species. The description presented is tedious for the reader.
The whole paragraph including Fig. 5 is not a discussion but still a description of the results. Besides, it is stated here how figure 4 was created. illogical layout of the paper
Line 277 unclear what is "habitat heterogeneity"? This statement is disconnected from the next sentence. 
Lines 284-289 another repetition of the observation but no discussion of why this situation occurs
Line 294 where there is confirmation of this sentence
Lines 353-354 the sentences are not corroborated in any way. Strongly enigmatic statements and not understandable
Line 368 what are ecological services? I am very sorry, but the content of the work contains so many unconnected threads that it is difficult to find your way around
Line 371 the sentence is not clear
Line 386 what this "good governance" consists of. This sentence is highly debatable against the background of the text presented.
Line 417 What kind of ES are the authors referring to?

Round 2

Reviewer 1 Report

No additional comments

Author Response

Response to Reviewer 1 Comments

The introduction has been carefully revised.

Reviewer 4 Report

I appreciate the authors' contribution to improving the manuscript. The manuscript has improved significantly and requires only minor corrections. 
Please state what quality of images were used for analysis. This element should not be questionable as it indicates the use of remote sensing methods in the assessed work.
An explanation of the need for multiple measures of diversity should be included in the research methodology chapter.

Author Response

This manuscript is a resubmission of an earlier submission. The following is a list of the peer review reports and author responses from that submission.